# Non-asymptotic convergence bounds for Wasserstein approximation using point clouds

**Quentin Mérigot**
Université Paris-Saclay, CNRS,
Laboratoire de mathématiques d'Orsay,
91405, Orsay, France
Institut Universitaire de France

**Filippo Santambrogio**
Univ Lyon,
Université Claude Bernard Lyon 1, CNRS
UMR 5208, Institut Camille Jordan
F-69622 Villeurbanne
Institut Universitaire de France

**Clément Sarrazin**
Université Paris-Saclay, CNRS,
Laboratoire de mathématiques d'Orsay
91405, Orsay, France

## Abstract

Several issues in machine learning and inverse problems require to generate discrete data, as if sampled from a model probability distribution. A common way to do so relies on the construction of a uniform probability distribution over a set of $N$ points which minimizes the Wasserstein distance to the model distribution. This minimization problem, where the unknowns are the positions of the atoms, is non-convex. Yet, in most cases, a suitably adjusted version of Lloyd's algorithm — in which Voronoi cells are replaced by Power cells — leads to configurations with small Wasserstein error. This is surprising because, again, of the non-convex nature of the problem, as well as the existence of spurious critical points. We provide explicit estimates for the convergence of this Lloyd-type algorithm, starting from a cloud of points that are sufficiently far from each other. Our estimates are tight when the algorithm is initialized from an point cloud that is evenly distributed in the ambient space. Similar bounds can be deduced for the corresponding gradient descent. These bounds naturally lead to a modified Poliak-Łojasiewicz inequality for the Wasserstein distance cost, with an error term depending on the distances between Dirac masses in the discrete distribution.

## 1   Introduction

In recent years, the theory of optimal transport has been the source of stimulating ideas in machine learning and in inverse problems. Optimal transport can be used to define distances, called Wasserstein or earth-mover distances, between probability distributions over a metric space. These distances allows one to measure the closeness between a generated distribution and a model distribution, and they have been used with success as data attachment terms in inverse problems. Practically, it has been observed for several different inverse problems that replacing usual loss functions with Wasserstein distances tend to increase the basin of convergence of the methods towards a good solution of the problem, or even to convexify the landscape of the minimized energy [8, 7]. This good behaviour is not fully understood, but one may attribute it partly to the fact that the Wasserstein distances encodes the geometry of the underlying space. A notable use of Wasserstein distances in machine learning is in the field of generative adversarial networks, where one seeks to design a neural network able to produce random examples whose distribution is close to a prescribed model distribution [2].

35th Conference on Neural Information Processing Systems (NeurIPS 2021).

**Wasserstein distance and Wasserstein regression**   Given two probability distributions $\rho, \mu$ on $\mathbb{R}^d$, the Wasserstein distance of exponent $p$ between $\rho$ and $\mu$ is a way to measure the total cost of moving mass distribution described by $\rho$ to $\mu$, knowing that moving a unit mass from $x$ to $y$ costs $\|x - y\|^p$. Formally, it is defined as the value of an optimal transport problem between $\rho$ and $\mu$:

$$W_p(\rho, \mu) = \left( \min_{\pi \in \Pi(\rho, \mu)} \int \|x - y\|^p \mathrm{d}\pi(x, y) \right)^{1/p}, \tag{1}$$

where we minimize over the set $\Pi(\rho, \mu)$ of *transport plans* between $\rho$ and $\mu$, i.e. probability distributions over $\mathbb{R}^d \times \mathbb{R}^d$ with marginals $\rho$ and $\mu$. Standard references on the theory of optimal transport include books by Villani and by Santambrogio [21, 22, 20], while the computational and statistical aspects are discussed in a survey of Cuturi and Peyré [17].

In this article, we consider regression problems with respect to the Wasserstein metric, which can be put in the following form

$$\min_{\theta \in \Theta} \mathrm{W}_p^p(T_{\theta\#}\mu, \rho), \tag{2}$$

where $\mu$ is the *reference distribution*, a probability measure on $[0, 1]^\ell$, $\rho$ is the *model distribution*, a probability measure on $\mathbb{R}^d$, and where $T_\theta : [0, 1]^\ell \to \mathbb{R}^d$ is a family of maps indexed by a parameter $\theta \in \Theta$. In the previous formula, we also denoted $T_{\theta\#}\mu$ the image of the measure $\mu$ under the map $T_\theta$, also called *pushforward* of $\mu$ under $T_\theta$. This image measure is defined by $T_{\theta\#}\mu(B) := \mu(T_\theta^{-1}(B))$ for any measurable set $B$ in the codomain of $T_\theta$. In this work, we will concentrate on the quadratic Wasserstein distance $W_2$. Several problems related to the design of generative models can be put under the form (2), see for instance [9, 2]. Alternatively, in some occasions ([4] and initially [18]), extracting a uniform quantization can be used as an intermediate step in k-means clustering. Let us note briefly here that our estimates do not make assumptions on the absolute continuity of the sampled measure and are therefore valid in this fully discrete case. In any case, solving (2) numerically is challenging for several reasons, but in this article we will concentrate on one of them: the non-convexity of the Wasserstein distance under displacement of the measures.

**Non-convexity of the Wasserstein distance under displacements.**   It is well known that the Wasserstein distance is convex for the standard (linear) structure of the space of probability measures, meaning that if $\nu_0$ and $\nu_1$ are two probability measures and $\nu_t = (1 - t)\nu_0 + t\nu_1$, then the map $t \in [0, 1] \mapsto \mathrm{W}_p^p(\nu_t, \rho)$ is convex. Using a terminology from physics, we may say that the Wasserstein distance is convex for the *Eulerian* structure of the space of probability measures, e.g. when one interpolates linearly between the densities. However, in the regression problem (2), the perturbations are *Lagrangian* rather than Eulerian, in the sense that modifications of the parameter $\theta$ leads to a displacement of the support of the measure $T_{\theta\#}\mu$. This appears very clearly in particular when $\mu$ is the uniform measure over a set $X = (x_1, \ldots, x_N)$ of $N$ point in $[0, 1]^d$, i.e. $\mu = \delta_X$ with

$$\delta_X \stackrel{\text{def}}{=} \frac{1}{N} \sum_{i=1}^{N} \delta_{x_i}.$$

In this case $T_{\theta\#}\mu$ is the uniform measure over the set $T_\theta(X) = (T_\theta(x_1), \ldots, T_\theta(x_n))$, i.e. $T_{\theta\#}\mu = \delta_{T_\theta(X)}$. In this article, we will therefore be interested by the function

$$F_N : Y \in (\mathbb{R}^d)^N \mapsto \frac{1}{2} W_2^2 (\rho, \delta_Y). \tag{3}$$

This function $F_N$ is *not* convex, and actually exhibits (semi-)concavity properties. This has been observed first in [1] (Theorem 7.3.2), and is related to the positive curvature of the Wasserstein space. A precise statement in the context considered here may also be found as Proposition 21 in [15]. A practical consequence of the lack of convexity of $F_N$ is that critical points of $F_N$ are not necessarily global minimizers. It is actually easy to construct examples of families of critical points $Y_N$ of $F_N$ such that $F_N(Y_N)$ is bounded from below by a positive constant, while $\lim_{N \to \infty} \min F_N = 0$, so that the ratio between $F_N(Y_N)$ and $\min F_N$ is arbitrarily large as $N \to +\infty$. This can be done by concentrating the points $Y_N$ on lower-dimensional subspaces of $\mathbb{R}^d$, as in Remarks 2 and 3.

When applying gradient descent to the nonconvex optimization problem (2), it is in principle possible to end up on local minima corresponding to a high energy critical points of the Wasserstein distance,

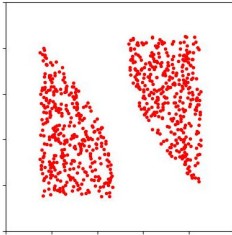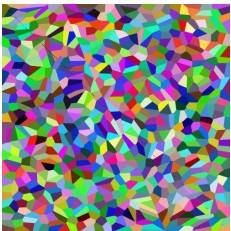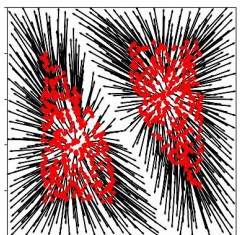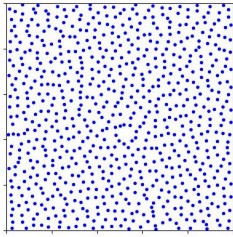

Figure 1: From left to right, a point cloud $Y^0$ in the square $\Omega = [0,1] \times [0,1]$, the associated power cells $P_i(Y)$ in the optimal transport to the Lebesgue measure on $\Omega$, the vectors $-N\nabla F_N(Y^0) = B_N(Y^0) - Y^0$ followed during the Lloyd step and the positions of the barycenters $Y^1 = B_N(Y)$.

regardless of the non-linearity of the map $\theta \mapsto T_\theta \# \mu$. Our main theorem, or rather its Corollary 6 shows that if the points of $Y$ are at distance at least $\varepsilon > 0$ from one another, then

$$F_N(Y) - C\frac{1}{N\varepsilon^{d-1}} \leq N\|\nabla F_N(Y)\|^2.$$

In the previous inequality, $\|\nabla F_N(Y)\|$ denotes the Euclidean norm of the vector in $\mathbb{R}^{Nd}$ obtained by putting one after the other the gradients of $F_N$ w.r.t. the positions of the atoms $y_i$. We note that due to the weights $1/N$ in the atomic measure $\delta_Y$, the components of this vector are in general of the order of $1/N$, see Proposition 1. This inequality resembles the Polyak-Łojasiewicz inequality, and shows in particular that if the quantization error $F_N(Y) = W_2^2(\rho, \delta_Y)$ is large, i.e. larger than $\varepsilon^{1-d}/N$, then the point cloud $Y$ is not critical for $F_N$. From this, we deduce in Theorem 7 that if the points in the initial cloud are not too close to each other at the initialization, then the iterates of fixed step gradient descent converge to points with low energy $F_N$, despite the non-convexity of $F_N$.

**Relation to optimal quantization**   Our main result also has implications in terms of the *uniform optimal quantization problem*, where one seeks a point cloud $Y = (y_1, \ldots, y_N)$ in $(\mathbb{R}^d)^N$ such that the uniform measure supported over $Y$, denoted $\delta_Y$, is as close as possible to the model distribution $\rho$ with respect to the 2-Wasserstein distance:

$$\min_{Y \in \Omega^N} F_N(Y). \tag{4}$$

The uniform optimal quantization problem (4) is a very natural variant of the (standard) *optimal quantization problem*, where one does not impose that the measure supported on $Y$ is uniform:

$$\min_{Y \in (\mathbb{R}^d)^N} G_N(Y), \quad \text{where } G_N : Y \in (\mathbb{R}^d)^N \mapsto \min_{\mu \in \Delta_N} W_2^2\left(\rho, \sum_{i=1}^{N} \mu_i \delta_{y_i}\right), \tag{5}$$

and where $\Delta_N \subseteq \mathbb{R}^N$ is the probability simplex. This standard optimal quantization problem is a cornerstone of sampling theory, and we refer the reader to the book of Graf and Luschgy [11] and to the survey by Pagès [16]. The uniform quantization problem (4) is less common, but also very natural. It has been used in imaging to produce stipplings of an image [5, 3] or for meshing purposes [10]. A common difficulty for solving (5) and (4) numerically is that the minimized functionals $F_N$ and $G_N$ are non-convex and have many critical points with high energy. However, in practice, simple fixed-point or gradient descent stategies behave well when the initial point cloud is not chosen adversely. Our second contribution is a quantitative explanation for this good behaviour in the case of the uniform optimal quantization problem.

Lloyd's algorithm [13] is a fixed point algorithm for solving approximately the standard optimal quantization problem (5). Starting from a point cloud $Y^k = (y_1^k, \ldots, y_N^k) \in (\mathbb{R}^d)^N$ with distinct points, one defines the next iterate $Y^{k+1}$ in two steps. First, one computes the Voronoi diagram of $Y$, a tessellation of the space into convex polyhedra $(V_i(Y^k))_{1 \leq i \leq N}$, where

$$V_i(Y) = \{x \in \Omega \mid \forall j \in \{1, \ldots, N\}, \|x - y_i\| \leq \|x - y_j\|\}. \tag{6}$$

In the second step, one moves every point $y_i^k$ towards the barycenter, with respect to $\rho$, of the corresponding cell $V_i(Y^k)$. This algorithm can also be interpreted as a fixed point algorithm for

solving the first-order optimality condition for (5), i.e. $\nabla G_N(Y) = 0$. One can show that the energy $(G_N(Y^k))_{k\geq 0}$ decreases in $k$. The convergence of $Y^k$ towards a critical point of $F_N$ as $k \to +\infty$ has been studied in [6], but the energy of this limit critical point is not guaranteed to be small.

In the case of the uniform quantization problem (4), one can try to minimize the energy $F_N$ by gradient descent, defining the iterates

$$Y^{k+1} = Y^k - \tau N \nabla F_N(Y^k), \tag{7}$$

where $\tau > 0$ is the time step. The factor $N$ in front of $\nabla F_N$ is set as a compensation for the fact that we have, in general, $\nabla F_N(Y) = O(1/N)$. When $\tau = 1$, one recovers a version of Lloyd's algorithm for the uniform quantization problem, involving barycenters $B_N(Y)$ of Power cells, rather than Voronoi cells, associated to $Y$. More precisely, Proposition 1 proves that $\nabla F_N(Y) = (Y - B_N(Y))/N$ so that $Y^{k+1} = B_N(Y^k)$ when $\tau = 1$. Quite surprisingly, we prove in Corollary 4 that if the points in the initial cloud $Y^0$ are not too close to each other, then the uniform measure over the point cloud $Y^1 = Y^0 - N \nabla F_N(Y^0)$ obtained after *only one step* of Lloyd's algorithm is close to $\rho$. This is illustrated in Figure 1. We prove in particular the following statement.

**Theorem** (Particular case of Corollary 4). *Let $\rho$ be a probability density over a compact convex set $\Omega \subseteq \mathbb{R}^d$, let $Y^0 = (y_1^0, \dots, y_N^0) \in \Omega^d$ and assume that the points lie at some positive distance from one another: for some constant $c$,*

$$\forall i \neq j, \|y_i^0 - y_j^0\| \geq cN^{-1/d},$$

*corresponding for instance to a point cloud sampled on a regular grid. Then, the point cloud $Y^1 = Y^0 - N \nabla F_N(Y^0)$ obtained after one step of Lloyd's algorithm satisfies*

$$W_2^2(\delta_{Y^1}, \rho) \leq C_{c,d,\Omega} N^{-1/d},$$

*where $C_{c,d,\Omega}$ is a constant depending on $c, d$ and $\mathrm{diam}(\Omega)$.*

**Outline**    In Section 2, we start by a short review of background material on optimal transport and optimal uniform quantization. We then establish our main result (Theorem 3) on the approximation of a measure $\rho$ by barycenters of Power cells. This theorem yields error estimates for one step of Lloyd's algorithm in deterministic and probabilistic settings (Corollaries 4 and 5). In Section 3, we establish a Polyak-Łojasiewicz-type inequality (Corollary 6) for the function $F_N : Y \mapsto \frac{1}{2} W_2^2(\rho, \delta_Y)$ introduced in (3), and we study the convergence of a gradient descent algorithm for $F_N$ (Theorem 7). Finally, in Section 4, we report numerical results on optimal uniform quantization in dimension $d = 2$.

## 2    Lloyd's algorithm for optimal uniform quantization

**Optimal transport and Kantorovich duality**    In this section we briefly review Kantorovich duality and its relation to semidiscrete optimal transport. The cost is fixed to $c(x, y) = \|x - y\|^2$, and we assume that $\rho$ is a probability density over a compact convex domain $\Omega$. In this setting, Brenier's theorem implies that given any probability measure $\mu$ supported on $\Omega$, the optimal transport plan between $\rho$ and $\mu$, i.e. the minimizer $\pi$ in the definition of the Wasserstein distance (1) with $p = 2$, is induced by a transport map $T_\mu : \Omega \to \Omega$, meaning $\pi = (T_\mu, Id)_\# \rho$. One can derive an alternative expression for the Wasserstein distance using Kantorovich duality, which leads to a more precise description of the optimal transport map [20, Theorem 1.39]:

$$W_2^2(\rho, \mu) = \max_{\phi:Y \to \mathbb{R}} \int_{\mathbb{R}^d} \phi \, \mathrm{d}\mu + \int_\Omega \phi^c \, \mathrm{d}\rho, \tag{8}$$

where $\phi^c(x) = \min_i c(x, y_i) - \phi_i$. When $\mu = \delta_Y$ is the uniform probability measure over a point cloud $Y = (y_1, \dots, y_N)$ containing $N$ distinct points, we set $\phi_i = \phi(y_i)$ and we define the *ith Power cell* associated to the couple $(Y, \phi)$ as

$$\mathrm{Pow}_i(Y, \phi) = \{x \in \mathbb{R}^d \mid \forall j \in \{1, \dots, N\}, \ \|x - y_i\|^2 - \phi_i \leq \|x - y_j\|^2 - \phi_j\}.$$

Then, the Kantorovich dual (8) of the optimal transport problem between $\rho$ and $\delta_Y$ turns into a finite-dimensional concave maximization problem

$$W_2^2(\mu, \rho) = \max_{\phi \in \mathbb{R}^N} \sum_{i=1}^N \frac{\phi_i}{N} + \int_{\mathrm{Pow}_i(Y, \phi)} \left( \|x - y_i\|^2 - \phi_i \right) \mathrm{d}\rho(x) \tag{9}$$

By Corollary 1.2 in [12], a vector $\phi \in \mathbb{R}^N$ is optimal for this maximization problem if and only if the potential $\phi$ is such that each Power cell contains the same amount of mass, i.e. if

$$\forall i \in \{1, \ldots, N\}, \ \rho(\mathrm{Pow}_i(Y, \phi)) = \frac{1}{N}, \tag{10}$$

From now on, we denote $P_i(Y) = \mathrm{Pow}_i(Y, \phi) \cap \Omega$, where $\phi \in \mathbb{R}^N$ satisfies (10). The optimal transport map $T_Y$ between $\rho$ and $\delta_Y$ sends every Power cell $P_i(Y)$ to the point $y_i$, i.e. it is defined $\rho$-almost everywhere by $T_Y|_{P_i(Y)} = y_i$. We refer again to the introduction of [12] for more details.

**Optimal uniform quantization**   In this article, we study the behaviour of the squared Wasserstein distance between the (fixed) probability density $\rho$ and a uniform finitely supported measure $\delta_Y$ where $Y = (y_1, \ldots, y_N)$ is a cloud of $N$ points, in terms of variations of $Y$. As in equation (3), we denote $F_N = \frac{1}{2} \mathrm{W}_2^2(\rho, \cdot)$. Proposition 21 in [15] gives an expression for the gradient of $F$, and proves its semiconcavity. We recall that $F$ is called $\alpha$–semiconcave, with $\alpha \geq 0$, if the function $F - \frac{\alpha}{2} \| \cdot \|^2$ is concave. We denote $\mathbb{D}_N$ the generalized diagonal

$$\mathbb{D}_N = \{Y \in (\mathbb{R}^d)^N \mid \exists i \neq j \text{ s.t. } y_i = y_j\}.$$

**Proposition 1** (Gradient of $F_N$). *The function $F_N$ is $\frac{1}{N}$–semiconcave on $(\mathbb{R}^d)^N$ and is of class $\mathcal{C}^1$ on $(\mathbb{R}^d)^N \setminus \mathbb{D}_N$. In addition, one has*

$$\forall Y \in (\mathbb{R}^d)^N \setminus \mathbb{D}_N, \nabla F_N(Y) = \frac{1}{N}(Y - B_N(Y)), \text{ where } B_N(Y) = (b_1(Y), \ldots, b_N(Y)) \tag{11}$$

*and where $b_i(Y)$ is the barycenter of the $i$th power cell, i.e. $b_i(Y) = N \int_{P_i(Y)} x \mathrm{d}\rho(x)$.*

It is not difficult to prove that $F_N$ admits at least one minimizer, and that this minimizer $Y$ satisfies the first-order optimality condition $Y = B_N(Y)$. A point cloud that satisfies this condition is called *critical*.

*Remark* 1 (Upper bound on the minimum of $F_N$).   We note from [15, Proposition 12] that when $\rho$ is supported on a compact subset of $\mathbb{R}^d$, then

$$\min F_N = \min_{Y \in (\mathbb{R}^d)^N} \frac{1}{2} \mathrm{W}_2^2(\rho, \delta_Y) \lesssim \begin{cases} N^{-\frac{2}{d}} & \text{if } d > 2 \\ N^{-1} \log N & \text{if } d = 2 \\ N^{-1} & \text{if } d = 1. \end{cases} \tag{12}$$

*Remark* 2 (High energy critical points).   On the other hand, since $F_N$ is not convex, this first-order condition is not sufficient to have a minimizer of $F_N$. For instance, if $\rho \equiv 1$ on the unit square $\Omega = [0,1]^2$, one can check that the point cloud

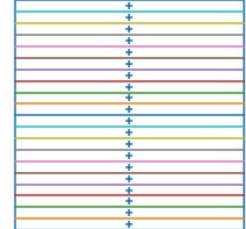

$$Y_N = \left( \left( \frac{1}{2N}, \frac{1}{2} \right), \left( \frac{3}{2N}, \frac{1}{2} \right), \ldots, \left( \frac{2N-1}{2N}, \frac{1}{2} \right) \right)$$

is a critical point of $F_N$ but not a minimizer of $F_N$. In fact, this critical point becomes arbitrarily bad as $N \to +\infty$ in the sense that

$$\lim_{N \to +\infty} \frac{F_N(Y_N)}{\min F_N} = +\infty.$$

On the other hand, we note that the point cloud $Y_N$ is highly concentrated, in the sense that the distance between consecutive points in $Y_N$ is $\frac{1}{2N}$, whereas in an evenly distributed point cloud, one would expect the minimum distance between points to be of order $N^{-1/d}$.

**Gradient descent and Lloyd's algorithm**   One can find a critical point of $F_N$ by following the discrete gradient flow of $F_N$, defined in (7), starting from an initial position $Y^0 \in (\mathbb{R}^d)^N \setminus \mathbb{D}_N$. Thanks to the expression of $\nabla F_N$ given in Proposition 1, the discrete gradient flow may be written as

$$Y^{k+1} = Y^k + \tau_N(B_N(Y^k) - Y^k), \tag{13}$$

where $\tau_N$ is a fixed time step. For $\tau_N = 1$, one recovers a variant of Lloyd's algorithm, where one moves every point to the barycenter of its Power cell $P_i(Y^k)$ at each iteration:

$$Y^{k+1} = B_N(Y^k) \tag{14}$$

We can state the following result about Lloyd's algorithm for the uniform quantization problem, whose proof is postponed to the appendix.

**Proposition 2.** *Let $N$ be a fixed integer and $(Y^k)_{k \geq 0}$ be the iterates of (14), with $Y^0 \notin \mathbb{D}_N$. Then, the energy $k \mapsto F_N(Y^k)$ is decreasing, and $\lim_{k \to +\infty} \|\nabla F_N(Y^k)\| = 0$. Moreover, the sequence $(Y^k)_{k \geq 0}$ belongs to a compact subset of $(\mathbb{R}^d)^N \setminus \mathbb{D}_N$ and every limit point of a converging subsequence of it is a critical point for $F_N$.*

Experiments suggest that following the discrete gradient flow of $F_N$ does not bring us to high energy critical points of $F_N$, such as those described in Remark 2, unless we started from an adversely chosen point cloud. The following theorem and its corollaries, the main results of this article, backs up this experimental evidence. It shows that if the point cloud $Y$ is not too concentrated, then the uniform measure over the barycenters of the power cells, $\delta_{B_N(Y)}$, is a good quantization of the probability density $\rho$, i.e. it bounds the quantization error after one step of Lloyd's algorithm (14).

We will use the following notation for $\varepsilon > 0$:

$$I_\varepsilon(Y) = \{i \in \{1, \ldots, N\} \mid \forall j \neq i, \|y_i - y_j\| \geq \varepsilon\}.$$

$$\mathbb{D}_{N,\varepsilon} = \{Y \in (\mathbb{R}^N)^d \mid \exists i \neq j, \|y_i - y_j\| \leq \varepsilon\}.$$

Note that $\mathbb{D}_{N,\varepsilon}$ is an $\varepsilon$-neighborhood around the generalized diagonal $\mathbb{D}_N$.

**Theorem 3** (Quantization by barycenters). *Let $\Omega \subseteq \mathbb{R}^d$ be a compact convex set, $\rho$ a probability density on $\Omega$ and consider a point cloud $Y = (y_1, \ldots, y_N)$ in $\Omega^N \setminus \mathbb{D}_N$. Then, for all $0 < \varepsilon \leq 1$,*

$$\mathrm{W}_2^2\left(\rho, \delta_{B_N(Y)}\right) \leq C_{d,\Omega} \left(\frac{\varepsilon^{1-d}}{N} + 1 - \frac{\mathrm{Card}(I_\varepsilon(Y))}{N}\right). \tag{15}$$

*where $C_{d,\Omega} = \frac{2^{2d-1}}{\omega_{d-1}}(\mathrm{diam}(\Omega) + 1)^{d+1}$ and where $\omega_{d-1}$ is the volume of the unit ball in $\mathbb{R}^{d-1}$.*

The proof relies on arguments from convex geometry. In particular, we denote $A \oplus B$ the Minkowski sum of sets: $A \oplus B = \{a + b \mid (a,b) \in A \times B\}$.

*Proof.* Let $\phi^1 \in \mathbb{R}^N$ be the solution to the dual Kantorovich problem (10) between $\rho$ and $\delta_Y$. We let $\phi^t = t\phi^1$ and we denote $P_i^t = \mathrm{Pow}_i(Y, \phi^t) \cap \Omega'$ the $i$th Power cell intersected with the slightly enlarged convex set $\Omega' = \Omega \oplus \mathrm{B}(0,1)$. This way, $P_i^1 \supseteq P_i(Y)$ whereas $P_i^0$ is in fact the intersection of the $i$-th Voronoi cell defined in (6) with $\Omega'$.

We will now prove an upper bound on the sum of the diameters of the cells $P_i(Y)$ whose index lies in $I_\varepsilon(Y)$. First, we notice the following inclusion, which holds for any $t \in [0,1]$:

$$(1-t)P_i^0 \oplus tP_i^1 \subseteq P_i^t, \tag{16}$$

Indeed, let $x^0 \in P_i^0$ and $x^1 \in P_i^1$, so that for all $j \in \{1, \ldots, N\}$ and $k \in \{0, 1\}$,

$$\|x^k - y_i\|^2 - \phi_i \leq \|x^k - y_j\|^2 - \phi_j.$$

Expanding the squares and substracting $\|x^k\|^2$ on both sides these inequalities become *linear* in $\phi_i, \phi_j$ and $x^k$, implying directly that $x^t = (1-t)x^0 + tx^1 \subseteq P_i^t$ as desired.

For any index $i \in I_\varepsilon$, the point $y_i$ is at distance at least $\varepsilon$ from other points, implying that $\mathrm{B}(0, \frac{\varepsilon}{2})$ is contained in the Voronoi cell $V_i(Y)$ with $\Omega'$. Using that $P_i^0 = V_i(Y) \cap \Omega'$, that $\Omega' = \Omega \oplus \mathrm{B}(0,1)$ and that $y_i \in \Omega$, we deduce that $P_i^0$ contains the same ball. On the other hand, $P_i^1$ contains a segment $S_i$ of length $\mathrm{diam}(P_i^1)$ and inclusion (16) with $t = \frac{1}{2}$ gives

$$\frac{1}{2}(\mathrm{B}(y_i, \varepsilon/2) \oplus S_i) \subseteq P_i^{1/2}.$$

The Minkowski sum in the left-hand side contains in particular the product of a $(d-1)$-dimensional ball of radius $\varepsilon/2$ with an orthogonal segment with length $\mathrm{diam}(P_i^1) \geq \mathrm{diam}(P_i(Y))$. Thus,

$$\frac{1}{2^d}\left(\omega_{d-1}\frac{\varepsilon^{d-1}}{2^{d-1}}\mathrm{diam}(P_i(Y))\right) \leq |P_i^{\frac{1}{2}}|,$$

where $|.|$ denotes the Lebesgue measure. Using that the Power cells $P_i^{\frac{1}{2}}$ form a tesselation of the domain $\Omega'$, we therefore obtain

$$\sum_{i \in I_\varepsilon(Y)} \text{diam}(P_i(Y)) \leq \frac{2^{2d-1}}{\omega_{d-1}} |\Omega'| \varepsilon^{1-d} \leq \frac{2^{2d-1}}{\omega_{d-1}} (\text{diam}(\Omega) + 1)^d \varepsilon^{1-d} \tag{17}$$

We now estimate the transport cost between $\delta_B$ and the density $\rho$, where $B = B_N(Y)$. The transport cost due to the points whose indices do not belong to $I_\varepsilon(Y)$ can be bounded in a crude way by

$$\sum_{i \notin I_\varepsilon(Y)} \int_{P_i(Y)} \|x - b_i(Y)\|^2 \mathrm{d}\rho(x) \leq (1 - \frac{\text{Card } I_\varepsilon(Y)}{N}) \text{diam}(\Omega)^2.$$

Note that we used $\rho(P_i(Y)) = \frac{1}{N}$. On the other hand, the transport cost associated with indices in $I_\varepsilon(Y)$ can be bounded using (17) and $\text{diam}(P_i(Y)) \leq \text{diam}(\Omega)$:

$$\sum_{i \in I_\varepsilon(Y)} \int_{P_i(Y)} \|x - b_i(Y)\|^2 \mathrm{d}\rho(x) \leq \frac{1}{N} \sum_{i \in I_\varepsilon(Y)} \text{diam}(P_i(Y))^2$$

$$\leq \frac{1}{N} \text{diam}(\Omega) \sum_{i \in I_\varepsilon} \text{diam}(P_i(Y))$$

$$\leq \frac{2^{2d-1}}{\omega_{d-1}} (\text{diam}(\Omega) + 1)^{d+1} \frac{\varepsilon^{1-d}}{N}$$

In conclusion, we obtain the desired estimate:

$$\text{W}_2^2 \left( \rho, \delta_{B_N(Y)} \right) \leq \frac{2^{2d-1}}{\omega_{d-1}} (\text{diam}(\Omega) + 1)^{d+1} \frac{\varepsilon^{1-d}}{N} + \text{diam}(\Omega)^2 \left( 1 - \frac{\text{Card } I_\varepsilon}{N} \right). \qquad \square$$

This theorem could be extended *mutatis mutandis* to the case where $\rho$ is a general probability measure (i.e. not a density). However, this would imply some technical complications in the definition of the barycenters $b_i$ by introducing a disintegration of $\rho$ with respect to the transport plan $\pi$.

**Consequence for the uniform Lloyd's algorithm** (14)   In the next corollary, we assume that any pair of distinct points in $Y_N \in (\mathbb{R}^d)^N$ is bounded from below by $\varepsilon_N \geq C N^{-\beta}$, implying that $I_{\varepsilon_N}(Y_N) = N$. This corresponds to the value one could expect for a point set uniformly sampled from a set with Minkowski dimension $\beta$. When $\beta > d - 1$, the corollary asserts that one step of Lloyd's algorithm is enough to approximate $\rho$, in the sense that the uniform measure $\delta_{B_N(Y_N)}$ over the barycenters converges towards $\rho$ as $N \to +\infty$.

**Corollary 4** (Quantization by barycenters, asymptotic case). *Assume* $\varepsilon_N \geq C \cdot N^{-1/\beta}$ *with* $C, \beta > 0$. *Then, with* $\alpha = 1 - \frac{d-1}{\beta}$

$$\forall Y \in (\mathbb{R}^d)^N \setminus \mathbb{D}_{N, \varepsilon_N}, \quad \text{W}_2^2(\rho, \delta_{B_N(Y)}) \leq \frac{C_{d,\Omega}}{C^{d-1}} N^{-\alpha}, \tag{18}$$

*and in particular, if* $\beta > d - 1$,

$$\lim_{N \to +\infty} \max_{Y \in (\mathbb{R}^d)^N \setminus \mathbb{D}_{\varepsilon_N}} \text{W}_2^2(\rho, \delta_{B_N(Y)}) = 0. \tag{19}$$

*Remark* 3 (Optimality of the exponent when $\beta = d$). There is no reason to believe that the exponent in the upper bound (18) is optimal in general. However, it seems to be optimal in a "worst-case sense" when $\beta = d$. More precisely, we show the following result in Appendix E: for any $\delta \in (0, 1)$, and for every $N = n^d$ ($n \in \mathbb{N}$) there exists a sequence of separable probability densities $\rho_N$ over $X = [-1, 1]^d$ ($\rho_N$ is a truncated Gaussian distributions, whose variance converges to zero slowly as $N \to +\infty$) such that if $Y_N$ is a uniform grid of size $n \times \cdots \times n = N^d$ in $X$, then

$$\text{W}_2^2(\delta_{B_N(Y_N)}, \rho_N) \geq C N^{-\frac{(2-\delta)}{d}},$$

where $C$ is independent of $N$. On the other hand, in this setting every point in $Y_N$ is at distance at least $C N^{-1/d}$ from any other point in $Y_N$. Applying Corollary 4 with $\beta = d$, i.e. $\alpha = \frac{1}{d}$, we get

$$\text{W}_2^2(\delta_{B_N(Y_N)}, \rho_N) \leq C' N^{-\frac{1}{d}}.$$

Comparing this upper bound on $\text{W}_2^2(\delta_{B_N(Y_N)}, \rho_N)$ with the above lower bound, one sees that is is not possible to improve the exponent.

*Remark* 4 (Optimality of (19)). The assumption $\beta > d - 1$ for (19) is tight: if $\rho$ is the Lebesgue measure on $[0,1]^d$, it is possible to construct a point cloud $Y_N$ with $N$ points on the $(d-1)$-cube $\{\frac{1}{2}\} \times [0,1]^{d-1}$ such that distinct point in $Y_N$ are at distance at least $\varepsilon_N \geq C \cdot N^{-1/(d-1)}$. Then, the barycenters $B_N(Y_N)$ are also contained in the cube, so that $W_2^2(\rho, \delta_{B_N(Y_N)}) \geq \frac{1}{12}$.

The next corollary is a probabilistic analogue of Corollary 4, assuming that the initial point cloud $Y$ is drawn from a probability density $\sigma$ on $\Omega$. Note that $\sigma$ can be distinct from $\rho$. The proof of this corollary relies on McDiarmid's inequality to quantify the proportion of $\varepsilon$-isolated points in a point cloud that is drawn randomly and independently from $\sigma$. The proof of this result is in Appendix B.

**Corollary 5** (Quantization by barycenters, probabilistic case). *Let $\sigma \in \mathrm{L}^\infty(\Omega)$ and let $X_1, ..., X_N$ be i.i.d. random variables with distribution $\sigma \in \mathrm{L}^\infty(\mathbb{R}^d)$. Then, there exists a constant $C > 0$ depending only on $\|\sigma\|_{\mathrm{L}^\infty}$ and $d$, such that for $N$ large enough,*

$$\mathbb{P}\left( W_2^2\left( \frac{1}{N}\sum_{i=1}^N \delta_{b_i^X}, \rho \right) \lesssim N^{-\frac{1}{2d-1}} \right) \geq 1 - e^{-CN^{\frac{2d-3}{2d-1}}}$$

## 3 Gradient flow and a Polyak-Łojasiewicz-type inequality

Theorem 3 can be interpreted as a modified Polyak-Łojasiewicz-type (PŁ for short) inequality for the function $F_N$. The usual PŁ inequality for a differentiable function $F : \mathbb{R}^D \to \mathbb{R}$ is of the form

$$\forall Y \in \mathbb{R}^D, \quad F(Y) - \min F \leq C\|\nabla F(Y)\|^2,$$

where $C$ is a positive constant. This inequality has been originally used by Polyak to prove convergence of gradient descent towards the global minimum of $F$. Note in particular that such an inequality implies that any critical point of $F$ is a global minimum of $F$. By Remark 2, $F_N$ has critical points that are not minimizers, so that we cannot expect the standard PŁ inequality to hold. What we get is a similar inequality relating $F_N(Y)$ and $\|\nabla F_N(Y)\|^2$ but with a term involving the minimimum distance between the points in place of $\min F_N$.

**Corollary 6** (Polyak-Łojasiewicz-type inequality). *Let $Y \in (\mathbb{R}^d)^N \setminus \mathbb{D}_{N,\varepsilon}$. Then,*

$$F_N(Y) - C_{d,\Omega}\frac{1}{N}\left(\frac{1}{\varepsilon}\right)^{d-1} \leq N\|\nabla F_N(Y)\|^2 \tag{20}$$

A proof of Corollary 6 can be found in Appendix C. We note that when $\varepsilon \simeq N^{-1/d}$, the second term of the left-hand side of (20) has order $N^{-1/d}$. On the other hand, as recalled in Remark 1, $\min F_N \lesssim N^{-2/d}$ when $d > 2$. Thus, (20) is strictly weaker than the PŁ inequality, which would involve the minimum of $F_N$.

**Convergence of a discrete gradient flow** The modified Polyak-Łojasiewicz inequality (20) suggests that the discrete gradient flow 13 will bring us close to a point cloud with low Wasserstein distance to $\rho$, provided that we can guarantee that the the points clouds $Y^k$ remain far from the generalized diagonal $\mathbb{D}_N$ during the iterations. We prove in Lemma 3 in Appendix D that if $Y^{k+1} = Y^k - \tau_N \nabla F_N(Y^k)$ and $\tau_N \in (0,1)$, then

$$\forall i \neq j, \quad \|y_i^{k+1} - y_j^{k+1}\| \geq (1 - \tau_N)\|y_i^k - y_j^k\|. \tag{21}$$

We note that this inequality ensures that $Y^k$ never touches the generalized diagonal $\mathbb{D}_N$, so that the gradient $\nabla F_N(Y^k)$ is well-defined at each step. Combining this inequality with Theorem 3, one can actually prove that if the points in the initial cloud $Y_N^0$ are not too close to each other, then a few steps of gradient discrete gradient descent leads to a discrete measure $Y_N^k$ that is close to the target $\rho$. We stress here that the goal of this paragraph is to showcase these kinds of several-steps estimates, in the simple case of our gradient flow for $F_N$. However, in this case, it provides a worse convergence rate to 0 (w.r.t. $N$) than that given by Corollary 4 and the decrease of the energy along the iterations from Proposition 2. Precisely, we arrive at the following theorem, proved in Appendix D.

**Theorem 7.** *Let $0 < \alpha < \frac{1}{d-1} - \frac{1}{d}$, $\varepsilon_N \gtrsim N^{-\frac{1}{d}-\alpha}$, and $Y_N^0 \in \Omega^N \setminus \mathbb{D}_{\varepsilon_N}$. Let $(Y_N^k)_k$ be the iterates of (13) starting from $Y_N^0$ with timestep $0 < \tau_N < 1$. We assume that $\lim_{N\to\infty} \tau_N = 0$ and we set*

$$k_N = \left\lfloor \frac{1}{d\tau_N}\ln(F_N(Y_N^0)N\varepsilon_N^{d-1}) \right\rfloor.$$

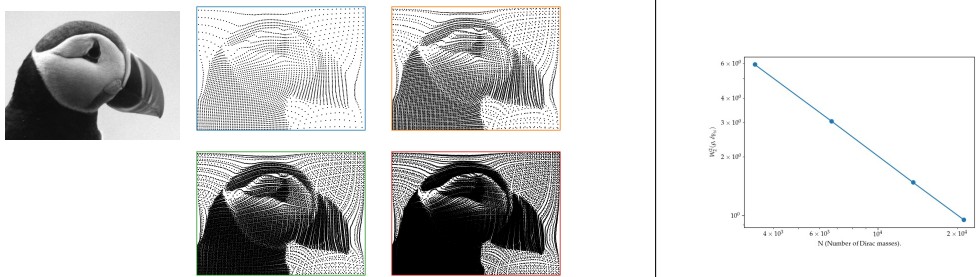

Figure 2: Optimal quantization of a density $\rho$ corresponding to a gray-scale image (Wikimedia Commons, CC BY-SA 3.0). (Left) We display the point clouds obtained after one step of Lloyd's algorithm, starting from a regular grid of size $N \in \{3750, 7350, 15000, 43350\}$. (Right) Quantization error $W_2^2\left(\rho, \delta_{B_N}\right)$ as a function of N the number of points, showing that $W_2^2\left(\rho, \delta_{B_N}\right) \simeq N^{-1.00}$.

*Then,*

$$W_2^2\left(\rho, \delta_{Y_N^{k_N}}\right) = O_{N \to \infty}\left(F_N(Y_N^0)^{1-\frac{1}{d}} \cdot N^{\frac{-1}{d^2}+\alpha\left(1-\frac{1}{d}\right)}\right). \tag{22}$$

*Remark* 5. Note that the exponential behavior implied by 21 and Lemma 3 is coherent with the estimates that are known in the absolutely continuous setting for the continuous gradient flow. When transitioning from discrete measures to probability densities, lower bounds on the distance between points become upper bounds on the density. The gradient flow $\dot{\mu}_t = \frac{1}{2}\nabla_\mu W_2^2(\rho, \mu_t)$ has an explicit solution $\mu_t = \sigma_{1-e^{-t}}$, where $\sigma$ is a constant-speed geodesic in the Wasserstein space with $\sigma_0 = \mu_0$ and $\sigma_1 = \rho$. In this case, a simple adaptation of the estimates in Theorem 2 in [19] shows the bound $\|\mu_t\|_{L^\infty} \le e^{td}\|\mu_0\|_{L^\infty}$. Still in this absolutely continous setting, it is possible to remove the exponential growth if the target density is also bounded, as a consequence of *displacement convexity* [14, Theorem 2.2]. There seems to be no discrete counterpart to this argument, explaining in part the discrepancy between the exponent of $N$ in (22) with the one obtained in Corollary 4.

## 4   Numerical results

In this section, we report some experimental results in dimension $d = 2$.

**Gray-scale image**   As we mentioned in the introduction, uniform optimal quantization allows to sparsely represent a (gray scale) image via points, clustered more closely in areas where the image is darker [5, 3]. On figure 2, we ploted the point clouds obtained after a single Lloyd step toward the density representing the image on the left (Puffin), starting from regular grids. The rate of convergence observed on the right-hand side chart, namely $N^{-1.00}$, is coherent with the theoretical estimate $\log(N)/N$ of Remark 1.

**Gaussian density with small variance**   We now consider a toy model where we approximate a gaussian density truncated to the unit square $\Omega = [0, 1]^2$, $\rho(x, y) = \frac{1}{Z}e^{-8\left((x-\frac{1}{2})^2+(y-\frac{1}{2})^2\right)}$ where $Z$ is a normalization constant. On the left column of Fig. 3, the initial point clouds $Y_N^0$ are randomly distributed in $[0, 1]^2$. The three point clouds represented above are obtained after one step of Lloyd's algorithm (14). The blue curve displays in a log-log scale the mean values of $F_N(B_N(Y_N))$ over a hundred random point clouds, for $N \in \{400, 961, 1600, 2500\}$. In this case, we observe a decrease rate $N^{-0.95}$ with respect to the number of points, similar to the case of the gray scale picture.

However, an interesting phenomena occurs when the initial point cloud $Y_N^0$ is aligned on a axis-aligned grid. The pictures in the right column of Fig. 3 where computed starting from such a grid with $N \in \{400, 961, 1600, 2500\}$ points. As in the randomly initialized case, we represented the values of $F_N(B_N(Y_N))$ in log-log scale. The corresponding discrete probability measure $\delta_{B_N(Y_N)}$ seems to converge to $\rho$ as $N \to \infty$, but with a much worse rate for these "low" values of $N$: $F_N(B_N(Y_N)) \simeq N^{-0.8}$. In this specific setting, with a separable density and an axis-aligned grid $Y_0$, the power cells are rectangles and a single Lloyd step brings us to a critical point of $F_N$. Thanks to this remark, it is possible to estimate the approximation error from the one-dimensional case. In fact, Appendix E shows that for any $\delta \in (0, 1)$, there exists variances $\sigma_N = \sigma_N(\delta)$ such that

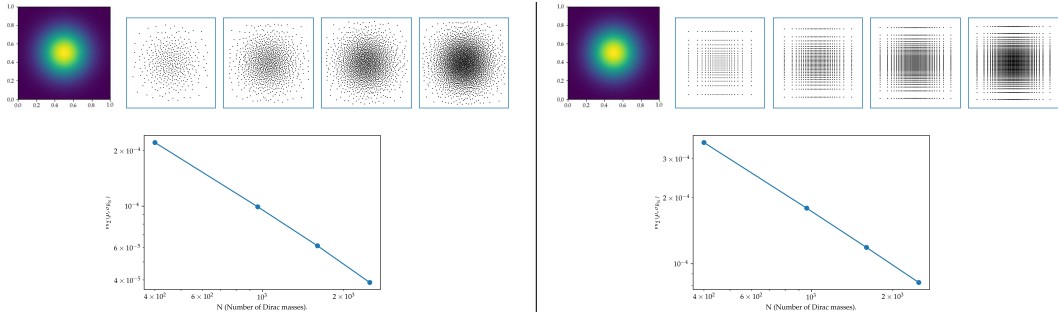

Figure 3: Optimal quantization of a Gaussian truncated to the unit square. On the left, the initial point cloud $Y_N$ is drawn randomly and *uniformly* from $[0,1]^2$, while on the right $Y_N$ is on a regular grid. The top row displays the point clouds obtained after one step of Lloyd's algorithm. The bottom row displays the quantization error after one step of Lloyd's algorithm $F_N(B_N(Y_N))$ as a fuction of the number of points. We get $F_N(B_N(Y_N)) \simeq N^{-0.95}$ when $Y_N$ is a random uniform point cloud in $[0,1]^N$ and $F_N(B_N(Y_N)) \simeq N^{-0.8}$ when $Y_N$ is a regular grid.

the approximation error $W_2^2(\rho_{\sigma_N}, \delta_{B_N})$ is of order $N^{-\frac{2-\delta}{2}}$. On the other hand, for a fixed $\sigma$, the approximation error is of order $N^{-1}$, to be compared with the bound $\log(N)/N$ for general measures.

## 5 Discussion

We have studied the problem of minimizing the Wasserstein distance between a fixed probability measure $\rho$ and a uniform measure over $N$ points $\delta_Y$, parametrized by the position of the points $Y = (y_1, \ldots, y_N)$. The main difficulty is the nonconvexity of the Wasserstein distance $F_N : Y \in (\mathbb{R}^d)^N \mapsto \frac{1}{2}W_2^2(\rho, \delta_Y)$, which we tackled by means of a modified Polyak-Łojaciewicz inequality (20). One limitation of our work is that the exponents in our estimates depends on the ambient dimension: it would be more interesting (but quite nontrivial) to establish similar estimates depending on the dimension of the support of $\rho$. Another direction of research would be to derive consequences for the algorithmic resolution of Wasserstein regression problems $\min_\theta W_2^2(\rho, T_{\theta\#}\mu)$, starting with the case where $\theta \mapsto T_\theta$ is linear.

## Acknowledgments and Disclosure of Funding

This work was supported by a grant from the French ANR (MAGA, ANR-16-CE40-0014).

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
