# A Proof of Proposition 2

Given $Y = (y_1, \ldots, y_N) \in (\mathbb{R}^d)^N \setminus \mathbb{D}_N$, one has for any $i \in \{1, \ldots, N\}$,

$$\int_{P_i(Y)} \|x - y_i\|^2 \, \mathrm{d}\rho(x) = \int_{P_i(Y)} \|x - b_i(Y) + b_i(Y) - y_i\|^2 \, \mathrm{d}\rho(x)$$

$$= \int_{P_i(Y)} \|x - b_i(Y)\|^2 \, \mathrm{d}\rho(x) + \frac{1}{N} \|b_i(Y) - y_i\|^2.$$

Summing these equalities over $i$ and remarking that the map $T_Y$ defined by $T_Y|_{P_i(Y)} = y_i$ is an optimal transport map between $\rho$ and $\delta_Y$, we get

$$\frac{1}{N} \|B_N(Y) - Y\|^2 = W_2^2(\rho, y_i) - \sum_i \int_{P_i(Y)} \|x - b_i(Y)\|^2 \, \mathrm{d}\rho(x)$$

$$\leq W_2^2(\rho, \delta_Y) - W_2^2(\rho, \delta_{B_N(Y)}).$$

Thus, with $Y^{k+1} = B_N(Y^k)$, we have

$$N \|\nabla F_N(Y^k)\|^2 = \frac{1}{N} \|Y^{k+1} - Y^k\|^2 \leq 2(F_N(Y^k) - F_N(Y^{k+1})).$$

This implies that the values of $F_N(Y^k)$ are decreasing in $k$ and, since they are bounded from below, that $\|\nabla F_N(Y^k)\| \to 0$ since $\sum_k \|\nabla F_N(Y^k)\|^2 < +\infty$. The sequence $(Y^k)_k$ can be easily seen to be bounded, since $F_N(Y^k)$ is bounded, which implies a bound on the second moment of $\delta_{Y^k}$.

For fixed $N$, since all atoms of $\delta_{Y^k}$ have mass $1/N$, this implies that all points $y_i^k$ belong to a same fixed compact ball. If $\rho$ itself is compactly supported, we can also prove that all points $Y^{k+1} = B_N(Y^k)$ are contained in a compact subset of $(\mathbb{R}^d)^N \setminus \mathbb{D}_N$, which means obtaining a lower bound on the distances $|b_i(Y) - b_j(Y)|$ for arbitrary $Y$. This lower bound can be obtained in the following way: since $\rho$ is absolutely continuous it is uniformly integrable which means that for every $\varepsilon > 0$ there is $\delta = \delta(\varepsilon) > 0$ such that for any set $A$ with Lebesgue measure $|A| < \delta$ we have $\rho(A) < \varepsilon$. We claim that we have $|b_i(Y) - b_j(Y)| \geq r := (2R)^{1-d} \delta(\frac{1}{2N})$, where $R$ is such that $\rho$ is supported in a ball $B_R$ of radius $R$. Indeed, it is enough to prove that every barycenter $b_i(Y)$ is at distance at least $r/2$ from each face of the convex polytope $P_i(Y)$. Consider a face of such a polytope and suppose, by simplicity, that it lies on the hyperplane $\{x_d = 0\}$ with the cell contained in $\{x_d \geq 0\}$. Let $s$ be such that $\rho(P_i(Y) \cap \{x_d > s\}) = \rho(P_i(Y) \cap \{x_d < s\}) = \frac{1}{2N}$. Then since the diameter of $P_i(Y) \cap B_R$ is smaller than $2R$, the Lebesgue measure of $P_i(Y) \cap \{x_d < s\}$ is bounded by $(2R)^{d-1} s$, which provides $s \geq r$ because of the definition of $r$. Since at least half of the mass (according to $\rho$) of the cell $P_i(Y)$ is above the level $x_d = s$ the $x_d$-coordinate of the barycenter is at least $r/2$. This shows that the barycenter lies at distance at least $r/2$ from each of its faces.

As a consequence, the iterations $Y^k$ of the Lloyd algorithm lie in a compact subset of $(\mathbb{R}^d)^N \setminus \mathbb{D}_N$, on which $F_N$ is $C^1$. This implies that any limit point must be a critical point.

We do not discuss here whether the whole sequence converges or not, which seems to be a delicate matter even for fixed $N$. It is anyway possible to prove (but we do not develop the details here) that the set of limit points is a closed connected subet of $(\mathbb{R}^d)^N$ with empty interior, composed of critical points of $F_N$ all lying on a same level set of $F_N$.

# B Proof of Corollary 5

Given $Y = (y_1, \ldots, y_N) \in (\mathbb{R}^d)^N$, we denote

$$I_\varepsilon(Y) = \{i \in \{1, \ldots, N\} \mid \forall j \neq i, \|y_i - y_j\| \geq \varepsilon\}, \quad \kappa_\varepsilon(Y) = \frac{1}{N} \operatorname{Card}(I_\varepsilon(Y)).$$

We call points $y_i$ such that $i \in I_\varepsilon(Y)$ $\varepsilon$-isolated, and points $y_i$ such that $i \notin I_\varepsilon(Y)$ $\varepsilon$-connected. Thus, $\kappa_\varepsilon$ gives the proportion of $\varepsilon$-isolated points in a cloud.

**Lemma 1.** *Let $X_1, \ldots, X_N$ be independent, $\mathbb{R}^d$-valued, random variables. Then, there is a constant $C_d > 0$ such that*

$$\mathbb{P}(\{|\kappa_\varepsilon(X_1, \ldots, X_N) - \mathbb{E}(\kappa_\varepsilon)| \geq \eta\}) \leq \mathrm{e}^{-N\eta^2/C_d}.$$

*Proof.* This lemma is a consequence of McDiarmid's inequality. To apply this inequality, we need evaluate the amplitude of variation of the function $\kappa_\varepsilon$ along changes of one of the points $x_i$. Denote $c_d$ the maximum cardinal of a subset $S$ of the ball $B(0, \varepsilon)$ such that the distance between any distinct points in $S$ is at least $\varepsilon$. By a scaling argument, one can check that $c_d$ does not, in fact, depend on $\varepsilon$. To evaluate

$$|\kappa_\varepsilon(x_1, \ldots, x_i, \ldots, x_N) - \kappa_\varepsilon(x_1, \ldots, \tilde{x}_i, \ldots, x_N)|,$$

we first note that at most $c_d$ points may become $\varepsilon$-isolated when removing $x_i$. To prove this, we remark that if a point $x_j$ becomes $\varepsilon$-isolated when $x_i$ is removed, this means that $\|x_i - x_j\| \le \varepsilon$ and $\|x_j - x_k\| > \varepsilon$ for all $k \notin \{i, j\}$. The number of such $j$ is bounded by $c_d$. Symmetrically, there may be at most $c_d$ points becoming $\varepsilon$-connected under addition of $\hat{x}_i$. Finally, the point $x_i$ itself may change status from $\varepsilon$-isolated to $\varepsilon$-connected. To summarize, we obtain that with $C_d = 2c_d + 1$,

$$|\kappa_\varepsilon(x_1, \ldots, x_i, \ldots, x_N) - \kappa_\varepsilon(x_1, \ldots, \tilde{x}_i, \ldots, x_N)| \le \frac{1}{N} C_d.$$

The conclusion then directly follows from McDiarmid's inequality. □

**Lemma 2.** *Let $\sigma \in \mathrm{L}^\infty(\mathbb{R}^d)$ be a probability density and let $X_1, \ldots, X_N$ be i.i.d. random variables with distribution $\sigma$. Then,*

$$\mathbb{E}(\kappa_\varepsilon(X_1, \ldots, X_N)) \ge (1 - \|\sigma\|_{\mathrm{L}^\infty} \omega_d \varepsilon^d)^{N-1}.$$

*Proof.* The probability that a point $X_i$ belongs to the ball $B(X_j, \varepsilon)$ for some $j \ne i$ can be bounded from above by $\sigma(B(X_j, \varepsilon)) \le \|\sigma\|_{\mathrm{L}^\infty} \omega_d \varepsilon^d$, where $\omega_d$ is the volume of the $d$-dimensional unit ball. Thus, the probability that $X_i$ is $\varepsilon$-isolated is larger than

$$(1 - \|\sigma\|_{\mathrm{L}^\infty} \omega_d \varepsilon^d)^{N-1}.$$

We conclude by noting that

$$\mathbb{E}(\kappa_\varepsilon(X_1, \ldots, X_N)) = \frac{1}{N} \sum_{1 \le i \le N} \mathbb{P}(X_i \text{ is } \varepsilon\text{-isolated}). \qquad \square$$

*Proof of Corollary 5.* We apply the previous Lemma 2 with $\varepsilon_N = N^{-\frac{1}{\beta}}$ and $\beta = d - \frac{1}{2}$. The expectation of $\kappa_{\varepsilon_N}(X_1, \ldots, X_N)$ is lower bounded by:

$$\mathbb{E}(\kappa_{\varepsilon_N}(X_1, \ldots, X_N)) \ge \left(1 - N^{-\frac{d}{\beta}} \|\sigma\|_{\mathrm{L}^\infty} \omega_d\right)^{N-1}$$
$$\ge 1 - CN^{1-\frac{d}{\beta}}$$

for large $N$, since $\beta < d$. By Lemma 1, for any $\eta > 0$,

$$\mathbb{P}(\kappa_{\varepsilon_N}(X_1, \ldots, X_N) \ge 1 - CN^{1-\frac{d}{\beta}} - \eta) \ge 1 - e^{-KN\eta^2},$$

for constants $C, K > 0$ depending only on $\|\sigma\|_{\mathrm{L}^\infty}$ and $d$. We choose $\eta = N^{-\frac{1}{2d-1}}$, so that $\eta$ is of the same order as $N^{1-\frac{d}{\beta}}$ since $1 - \frac{d}{\beta} = -\frac{1}{2d-1}$. Thus, for a slightly different $C$,

$$\mathbb{P}(\kappa_{\varepsilon_N}(X_1, \ldots, X_N) \ge 1 - C\eta) \ge 1 - e^{-KN\eta^2}.$$

Now, for $\omega_1, \ldots, \omega_N$ such that

$$\kappa_{\varepsilon_N}(X_1(\omega_1), \ldots, X_N(\omega_N)) \ge 1 - C\eta,$$

Theorem 3 yields:

$$W_2^2\left(\delta_{B_N(X(\omega))}, \rho\right) \lesssim \frac{N^{\frac{d-1}{\beta}}}{N} + \eta \lesssim N^{-\frac{1}{2d-1}}$$

and such a disposition happens with probability at least

$$1 - e^{-KN\eta^2} = 1 - e^{-KN^{\frac{2d-3}{2d-1}}}. \qquad \square$$

# C Proof of Corollary 6

We first note that by Proposition 1, we have $\|\nabla F_N(Y)\|^2 = \frac{1}{N^2}\|B_N(Y) - Y\|^2$. We then use $W_2^2(\delta_{B_N(Y)}, \delta_Y) \le \frac{1}{N}\|B_N(Y) - Y\|^2$ and

$$W_2^2(\rho, \delta_Y) \le 2W_2^2(\rho, \delta_{B_N(Y)}) + 2N\|\nabla F_N(Y)\|^2.$$

Thus, using Theorem 3 to bound $W_2^2(\rho, \delta_{B_N(Y)})$ from above, we get the desired result.

# D Proof of Theorem 7

**Lemma 3.** *Let* $Y^0 \in (\mathbb{R}^d)^N \setminus \mathbb{D}_{N,\varepsilon_N}$ *for some* $\varepsilon_N > 0$. *Then, the iterates* $(Y^k)_{k \ge 0}$ *of (13) satisfy for every* $k \ge 0$, *and for every* $i \ne j$

$$\|y_i^k - y_j^k\| \ge (1 - \tau_N)^k \varepsilon_N \tag{23}$$

*Proof.* We consider the distance between two trajectories after $k$ iterations: $e_k = \|y_i^k - y_j^k\|$. Assuming that $e_k > 0$, the convexity of the norm immediately gives us:

$$e_{k+1} - e_k \ge \left(\frac{y_i^k - y_j^k}{\|y_i^k - y_j^k\|}\right) \cdot \left(y_i^{k+1} - y_j^{k+1} - (y_i^k - y_j^k)\right)$$

$$= \tau_N \left(\frac{y_i^k - y_j^k}{\|y_i^k - y_j^k\|}\right) \cdot \left(b_i^k - b_j^k\right) - \tau_N \|y_i^k - y_j^k\|$$

where we denoted $b_i^k := b_i(Y_N^k)$ the barycenter of the $i$th Power cell $P_i(Y_N^k)$ in the tesselation associated with the point cloud $Y_N^k$. Since each barycenter $b_i^k$ lies in its corresponding Power cell, the scalar product $\left(y_i^k - y_j^k\right) \cdot \left(b_i^k - b_j^k\right)$ is non-negative: Indeed, for any $i \ne j$,

$$\left\|y_i^k - b_i^k\right\|^2 - \left\|y_j^k - b_i^k\right\|^2 \le \phi_i^k - \phi_j^k$$

Summing this inequality with the same inequality with the roles of $i$ and $j$ reversed, we obtain:

$$\left(y_i^k - y_j^k\right) \cdot \left(b_i^k - b_j^k\right) \ge 0$$

thus giving us the geometric inequality $e_{k+1} \ge (1 - \tau_N)e_k$. Since $Y_N^0$ was chosen in $\Omega^N \setminus \mathbb{D}_{N,\varepsilon_N}$, this yields $e_k \ge (1 - \tau_N)^k e_0$ and inequality 23. $\square$

**Lemma 4.** *For any* $k \ge 0$

$$F_N(Y_N^k) \le F_N(Y_N^0)\eta_N^k + 2C_{d,\Omega}(1 - \eta_N)\frac{\varepsilon_N^{1-d}}{N}\frac{A_N^k - \eta_N^k}{A_N - \eta_N}, \tag{24}$$

*where we denote* $\eta_N = 1 - \frac{\tau_N}{2}(2 - \tau_N)$ *and* $A_N = (1 - \tau_N)^{1-d}$.

*Proof.* This is obtained in a very similar fashion as Lemma 3. For any $k \ge 0$, the semi-concavity of $F_N$ yields the inequality:

$$F_N(Y_N^{k+1}) - \frac{\|Y_N^{k+1}\|^2}{2N} - \left(F_N(Y_N^k) - \frac{\|Y_N^k\|^2}{2N}\right) \le \left(-\frac{B_N^k}{N}\right) \cdot \left(Y_N^{k+1} - Y_N^k\right)$$

with $B_N^k := B_N(Y_N^k)$ in accordance with the previous proof.

Rearranging the terms,

$$F_N(Y_N^{k+1}) - F_N(Y_N^k) \le -\tau_N(1 - \frac{\tau_N}{2})\frac{\left\|B_N^k - Y_N^k\right\|^2}{N}$$

$$= -\tau_N(1 - \frac{\tau_N}{2})W_2^2(\delta_{B_N^k}, \delta_{Y_N^k})$$

$$\le \tau_N(1 - \frac{\tau_N}{2})\left(-\frac{1}{2}W_2^2(\delta_{Y_N^k}, \rho) + W_2^2(\rho, \delta_{B_N^k})\right)$$

by applying first the triangle inequality to $W_2(\delta_{B_N^k}, \delta_{Y_N^k})$ and then Cauchy-Schwartz's inequality. Using Theorem 3, this yields:

$$F_N(Y_N^{k+1}) \leq (1 - \frac{\tau_N}{2}(2 - \tau_N))F_N(Y_N^k) + 2C_{d,\Omega}\tau_N(2 - \tau_N)\frac{\varepsilon_N^{1-d}}{N}(1 - \tau_N)^{k(1-d)}$$

$$\leq \eta_N F_N(Y_N^k) + 2C_{d,\Omega}(1 - \eta_N)\frac{\varepsilon_N^{1-d}}{N}A_N^k.$$

and we simply iterate on $k$ to end up with the bound claimed in Lemma 4. □

*Proof of Theorem 7.* To conclude, we simply make (order 1) expansions of the terms in 24. The definition of $k_N$ in Theorem 7, although convoluted, was made so that both terms in the right-hand side of this inequality, $F_N(Y_N^0)\eta_N^{k_N}$ and $(1 - \eta_N)\frac{\varepsilon_N^{1-d}}{N}\frac{A_N^{k_N} - \eta_N^{k_N}}{A_N - \eta_N}$ have the same asymptotic decay to 0 (as $N \to +\infty$): With the notations of the previous proposition, we have for fixed $N$:

$$W_2^2\left(\rho, \delta_{Y_N^{k_N}}\right) \leq W_2^2\left(\rho, \delta_{Y_N^0}\right)\eta_N^{k_N} + 2C_{d,\Omega}\frac{(1 - \eta_N)}{A_N - \eta_N}\frac{A_N^{k_N} - \eta_N^{k_N}}{N\varepsilon_N^{d-1}} \tag{25}$$

We make use here of the notation from Section 3:

$$T_N = k_N\tau_N = \left\lfloor\frac{1}{d}\ln(F_N(Y_N^0)N\varepsilon_N^{d-1})\right\rfloor$$

to clear this expression a bit, and, because of the assumption $\lim_{N\to\infty}\tau_N = 0$, we may write:

$$\frac{A_N^{k_N} - \eta_N^{k_N}}{N\varepsilon_N^{d-1}} = \frac{e^{(d-1)T_N}}{N\varepsilon_N^{d-1}} + o_{N\to\infty}\left(\frac{T_N}{(N\varepsilon_N^{d-1})^{\frac{1}{d}}}\right)$$

as well as $\eta_N^{k_N} = e^{-T_N} + o_{N\to\infty}\left(\frac{T_N}{(N\varepsilon_N^{d-1})^{\frac{1}{d}}}\right)$, and substituting $T_N$,

$$W_2^2\left(\rho, \delta_{Y_N^{k_N}}\right) \lesssim \frac{W_2^2\left(\rho, \delta_{Y_N^0}\right)^{\frac{d-1}{d}}}{\left(N\varepsilon_N^{d-1}\right)^{\frac{1}{d}}} + o_{N\to\infty}\left(\frac{T_N}{(N\varepsilon_N^{d-1})^{\frac{1}{d}}}\right)$$

$$\lesssim W_2^2\left(\rho, \delta_{Y_N^0}\right)^{1 - \frac{1}{d}}N^{\frac{-1}{d^2} + \alpha\left(1 - \frac{1}{d}\right)} \qquad □$$

# E  Case of a low variance Gaussian in Section 4

Here, we consider $\rho_\sigma$ the probability measure obtained by truncating and renormalizing a centered normal distribution with variance $\sigma$ to the segment $[-1, 1]$. We first show that for any $N \in \mathbb{N}$ and $\delta \in (0, 1)$, we can find a small $\sigma_{N,\delta}$ such that the Wasserstein distance beween $\rho_{\sigma_{N,\delta}}$ and its best $N$-points approximation of is at least $CN^{-(2-\delta)}$.

**Proposition 8.** *For any $\sigma > 0$, consider $\rho_\sigma \overset{def}{=} m_\sigma e^{-\frac{|x|^2}{2\sigma^2}}\mathbb{1}_{[-1;1]}dx$ the truncated centered Gaussian density, where $m_\sigma$ is taken so that $\rho_\sigma$ has unit mass. Then, for every $\delta \in (0, 1)$, there exists a constant $C > 0$ and a sequence of variances $(\sigma_N)_{N\in\mathbb{N}}$ such that*

$$\forall Y \in (\mathbb{R}^d)^N \setminus \mathbb{D}_N, \quad W_2^2\left(\delta_{B_N(Y)}, \rho_{\sigma_N}\right) \geq CN^{-(2-\delta)}$$

From the proof, one can see that the dependence of $\sigma_N$ on $N$ is logarithmic.

*Proof.* We denote $g : x \in \mathbb{R} \mapsto \frac{1}{\sqrt{2\pi}}e^{-\frac{|x|^2}{2}}$ the density of the centered Gaussian distribution and $F_g$ its cumulative distribution function, so that

$$m_\sigma^{-1} = \int_{-1}^1 e^{-\frac{|x|^2}{2\sigma^2}}dx = \sigma\sqrt{2\pi}\int_{-1/\sigma}^{1/\sigma}g(y)dy = \sqrt{2\pi}\sigma(F_g(1/\sigma) - F_g(-1/\sigma)) \tag{26}$$

Note that, whenever $\sigma \to 0$, we have $(\sigma m_\sigma)^{-1} \to \sqrt{2\pi}$. We denote by $F_\sigma : [-1, 1] \to [0, 1]$ the cumulative distribution function of $\rho_\sigma$. Given any point cloud $Y = (y_1, \ldots, y_N)$ such that $y_1 \leq \ldots \leq y_N$, the Power cells $P_i(Y)$ is simply the segment

$$P_i(Y) = [F_\sigma^{-1}(i/N), F_\sigma^{-1}((i+1)/N)].$$

Since these segments do not depend on $Y$, we will denote them $(P_i)_{1 \leq i \leq N}$. Finally, defining $b_i = N \int_{P_i} x \mathrm{d}\rho_\sigma(x)$ as the barycenter of the $i$th power cell and $\delta_B = \frac{1}{N} \sum_i \delta_{b_i}$, we have

$$
\begin{aligned}
W_2^2(\delta_B, \rho_\sigma) &= \sum_{i=1}^N \int_{P_i} (x - b_i)^2 \mathrm{d}\rho_\sigma(x) \\
&\geq \rho_\sigma(-1) \sum_{i=1}^N \int_{P_i} (x - b_i)^2 \mathrm{d}x \\
&\geq C\rho_\sigma(-1) \sum_{i=1}^N (F_\sigma^{-1}((i+1)/N) - F_\sigma^{-1}(i/N))^3,
\end{aligned}
\tag{27}
$$

where we used that $\rho_\sigma$ attains its minimum at $\pm 1$ to get the first inequality. We now wish to provide an approximation for $F_\sigma^{-1}(t)$, $t \in [0, 1]$. We first note, using Taylor's formula, that we have

$$
\begin{aligned}
F_\sigma^{-1}(t) &= \sigma F_g^{-1}\left(F_g\left(\frac{-1}{\sigma}\right) + t\left[F_g\left(\frac{1}{\sigma}\right) - F_g\left(\frac{-1}{\sigma}\right)\right]\right) \\
&= \sigma F_g^{-1}\left(F_g\left(\frac{-1}{\sigma}\right) + \frac{t}{\sqrt{2\pi}\sigma m_\sigma}\right) \\
&= -1 + \sigma(F_g^{-1})'\left(F_g\left(\frac{-1}{\sigma}\right)\right) \frac{t}{\sqrt{2\pi}\sigma m_\sigma} + \frac{\sigma}{2}(F_g^{-1})''(s) \frac{t^2}{2\pi\sigma^2 m_\sigma^2}
\end{aligned}
$$

for some $s \in [F_g(-\frac{1}{\sigma}), F_g(-\frac{1}{\sigma}) + t(F_g(\frac{1}{\sigma}) - F_g(-\frac{1}{\sigma}))]$. But,

$$(F_g^{-1})'(t) = \frac{1}{g \circ F_g^{-1}(t)} = \sqrt{2\pi} e^{\frac{|F_g^{-1}(t)|^2}{2}},$$

$$(F_g^{-1})''(t) = -\frac{g' \circ F_g^{-1}(t)}{\left(g \circ F_g^{-1}(t)\right)^3} = 2\pi F_g^{-1}(t) e^{|F_g^{-1}(t)|^2},$$

and we see that

$$\left| F_\sigma^{-1}(t) - \left(-1 + \frac{t}{m_\sigma} e^{\frac{1}{2\sigma^2}}\right) \right| \leq e^{\frac{1}{\sigma^2}} \frac{t^2}{2\sigma^2 m_\sigma^2}$$

Therefore, if we denote $\varepsilon(\sigma, t)$ the second-order error in the above formula, i.e. $\varepsilon(\sigma, t) = e^{\frac{1}{\sigma^2}} \frac{t^2}{2\sigma^2 m_\sigma^2}$, the size of the first Power cell $P_0(Y)$ is of order:

$$F_\sigma^{-1}(1/N) - F_\sigma^{-1}(0) = \frac{1}{Nm_\sigma} e^{\frac{1}{2\sigma^2}} + O\left(\varepsilon\left(\sigma, \frac{1}{N}\right)\right).$$

We will choose $\sigma_N$ depending on $N$ in order for the first term in the left-hand side to dominate the second one:

$$\varepsilon\left(\sigma_N, \frac{1}{N}\right) = o\left(\frac{1}{Nm_\sigma} e^{\frac{1}{2\sigma^2}}\right). \tag{28}$$

In this way, we have

$$
\begin{aligned}
(F_\sigma^{-1}(1/N) - F_\sigma^{-1}(0))^3 \rho_\sigma(-1) &\geq c \frac{1}{N^3 m_\sigma^3} e^{\frac{3}{2\sigma^2}} m_\sigma e^{-\frac{1}{2\sigma^2}} \\
&= c \frac{1}{N^3 m_\sigma^2} e^{\frac{1}{\sigma^2}}.
\end{aligned}
\tag{29}
$$

We now choose $\sigma = \sigma_N$ such that $e^{\frac{1}{2\sigma^2}} = N^\alpha$ for an exponent $\alpha$ to be chosen. We need $\alpha > 0$ so that $\sigma_N \to 0$. This last condition and (26) implies that $m_{\sigma_N}$ is of order $\sqrt{\log N}$. This means that the condition (28) is satisfied if $\alpha < 1$ and $N$ large enough.

The sum in (27) is lower bounded by its first term, (29), and we get

$$W_2^2(\delta_B, \rho_\sigma) \geq c \frac{1}{N^3 m_{\sigma_N}^2} e^{\frac{1}{\sigma_N^2}} \geq C \left( \frac{N^{2\alpha-3}}{\ln(N)} \right)$$

for some constant $C > 0$, since $\sigma$ depends logarithmically on $N$. Finally, if we want this last expression to be larger than $N^{-(2-\delta)}$ we can take for instance $2\alpha > 1 + \delta$ and $N$ large enough. $\qquad \square$

The following corollary, whose proof can just be obtained by adapting the above proof to a simple multi-dimensional setting where measures and cells "factorize" according to the components, confirms the facts observed in the numerical section (Section 4), and the sharpness of our result (Remark 4).

**Corollary 9.** *Fix $\delta \in (0, 1)$. Given any $n \in \mathbb{N}$, consider an axis-aligned discrete grid of the form $Z_N = Y_1 \times \ldots \times Y_d$ in $\mathbb{R}^d$, with $N = \mathrm{Card}(Z_N) = n^d$, where each $Y_j$ is a subset of $\mathbb{R}$ with cardinal $n$. Finally, define $\sigma_N := \sigma_{n,\delta}$ as in Proposition 8 Then we have*

$$W_2^2(\delta_{B_N(Z_N)}, \rho_{\sigma_N} \otimes \cdots \otimes \rho_{\sigma_N}) \geq CN^{-\frac{(2-\delta)}{d}},$$

*where the constant $C$ is independent of $N$.*