# OpenReview forum: "Non-asymptotic convergence bounds for Wasserstein approximation using point clouds"
_NeurIPS.cc/2021/Conference — NeurIPS 2021 Poster_

### Official Review · Reviewer_u2eX · 2021-07-03

**Rating:** 9
**Confidence:** 4

**Summary:**

Authors study a variant of the quantization problem: given a probability measure $\rho$ on $\mathbb{R}^d$, we are looking for the uniform measure on $N$ points that approximates the best $\rho$ for the Wasserstein distance. Even though this problem is non-convex, it is shown that one step of a gradient descent allows one to reach a configuration with low energy (of order $N^{-1/d}$, whereas the minimum is of order $N^{-2/d}$). Furthermore, computing the gradient boils down to performing one step of a variant of Lloyd's algorithm.

**Limitations And Societal Impact:**

Yes

**Main Review:**

COMMENTS:

The paper addresses a variant of the classical quantization problem, that is well-motivated as an instance of a Wasserstein regression problem. Understanding why the OT approach is efficient in ML is an important topic, and proving that a (non-convex) Wasserstein regression problem can be solved by a gradient descent participates to this general goal.
The main result (one step of a gradient descent is enough to attain a region with low energy) is nice and quite surprising.
 The overall presentation of the paper is clear, and the different implications of the main result (Theorem 3) are explored in detail. I also appreciate that the limitations of the paper are clearly laid out: in its current form, the main theorem asserts that one can attain a region with low energy, but this energy can still be far from being minimal. The theoretical results are also showcased on simple examples (Figure 1 being particularly striking).

QUESTIONS:

I would like the authors to explain in a few sentences whether the behavior shown in Theorem 3 also occurs for the classical quantization problem. Can similar proof techniques be applied to this different setting?

Also, I have trouble understanding Theorem 7. Indeed, the rate of convergence in this theorem is slower than $N^{-1/d}$, whereas this is the energy we attain with one step in Theorem 3. The hypothesis $\epsilon_N \gtrsim N^{-1/d-\alpha}$ is a little bit weaker than the assumption $\epsilon_N\gtrsim N^{-1/d}$ of Theorem 3, but still, it is weird that as $\alpha$ goes to 0 we do not recover a rate that is at least as good as in Theorem 3. Could the authors elaborate on that?


MINOR MISTAKES/TYPOS:

l.33: minimize OVER the set

l.54: examples OF families

l.60: $\theta \mapsto T_{\theta} \mu$

l.109: $F_N:Y\mapsto ...$

l.127: Power cell containS

l.132: ''for any Y\in D_n" should be removed.

l.168: $\phi_i^k$ in the equation above

l.170: second and third equation: I believe that the power cell should be transported towards $b_i(Y)$ and not $y_i$.

l.205: Mention that the proof is in the Appendix

l.209: provided that WE.. / far FROM THE

l.373: I_\eps(Y) already defined. \kappa should be defined

l.381: remove ( in the equation

l.405: A_N instead of A (also below)

l.416/417: We do not have \sigma m_\sigma \to \sqrt{2\pi}.

l.416/417: remove ..., \leq y_N)


**Time Spent Reviewing:**

7

---

> ### Author Response · Authors · 2021-08-10
> **Response to the first review**
>
> Dear Reviewer,
>
> We would like to start by thanking you for your thorough review and critique of our submission. We have tried to answer them in a satisfying fashion. Let us provide a more elaborate answer for some of them:
>
> >  I would like the authors to explain in a few sentences whether the behavior shown in Theorem 3 also occurs for the classical quantization problem. Can similar proof techniques be applied to this different setting?
>
> $\rightarrow$ Good question! The behaviour of Theorem 3 does not seem to hold in the classical quantization problem. Assume for instance that rho is uniform on two square joined by a tiny corridor orthogonal to one of the sides, and that $Y^0$ is contained only in one of the two squares. Then, after one step of the classical Lloyd's algorithm, only the points of the right-most "column" will have moved to the
> right-hand side, so that the Wasserstein distance will be of order $(d-1)/d$.
>
> > Also, I have trouble understanding Theorem 7. [...] Could the authors elaborate on that?
>
> $\rightarrow$ Theorem 7 does not give a better convergence rate for several steps of the Lloyd algorithm. The main issue here is that during this algorithm, the distance between two points could reduce sufficiently for the bound given by Theorem 3 to stop converging to 0 as $N$ goes to infinity.
>
> We will correct all the typos you highlighted, thank you for your thorough reading!

---

> > ### Comment · Reviewer_u2eX · 2021-08-20
> > **Theorem 7**
> >
> > Thank you for your response.
> >
> > It appears I am still misunderstanding Theorem 7. Indeed, as the rate of convergence of Theorem 7 is worse than the rate obtained after one step, I do not understand why this result is interesting. After all, one could choose $k_N=1$ and $\tau_N=1$, and we would obtain a faster convergence rate from Theorem 3. Why is the choice of $k_N$ proposed in Theorem 7 relevant?

---

> > > ### Author Response · Authors · 2021-08-24
> > > **Re: Theorem 7**
> > >
> > > We agree that setting $\tau_N = 1$ and $k_N = 1$ in Theorem 7 gives a worse estimation than the one obtained with the "one step" analysis (Theorem 3). However, we believe that the strategy used to prove Theorem 7, which compares the time-discretized gradient flow to its continuous counterpart, could be adapted to more general problems, such as those where the minimized functional is the sum of the squared Wasserstein distance and a regularization term, or perhaps even Wasserstein regression problems of the form $\min_{\theta} W_2(T_{\theta} \mu,\rho)$. This is why we chose to include Theorem 7 and its proof in the article.

---

> > > > ### Comment · Reviewer_u2eX · 2021-08-25
> > > > **Re: Theorem 7**
> > > >
> > > > Thank you for answering. I believe your response should be made more explicit in the final version. While rereading the paper, it does not appear clearly that we are not trying to obtain better rates of convergence in the last section. Arguments suggesting that this approach could help solving the more general problem would be welcome.
> > > >
> > > > Otherwise, I am convinced.

---

### Official Review · Reviewer_nnEq · 2021-07-12

**Rating:** 8
**Confidence:** 4

**Summary:**

This paper studies the non-convexity that arises in Optimal Transport problems when trying to optimize particle locations (a.k.a. Lagrangian approach), typically when optimizing maps $F_N : Y = (y_1,\dots, y_N) \mapsto W_2^2(\rho, \frac{1}{N} \sum_i \delta_{y_i})$ (_uniform_ quantization problem with reference measure $\rho$). The map $F_N$ admits spurious local minima $Y^\mathrm{bad}_N$ which in particular satisfy $F_N(Y^\mathrm{bad}_N) \not\to 0$ when $N \to \infty$. However, the paper proves that, given a proper (non-adversarial) initialization, a single step of (a variant of) the Lloyd algorithm is sufficient to obtain a quantization $Y_N$ that satisfy $F_N(Y_N) \to 0$ (with rate $\leq N^{-1/d}$). Authors then prove a ''``PL-type'' inequality and use it to prove convergence (with rates) of a gradient descent scheme to minimize $F_N$.

**Limitations And Societal Impact:**

The paper explicitly discusses some of its limitations, which is appreciated. I did not identified specific negative social impact the paper could have.

**Main Review:**

Clarity:
The paper is well-written and very pleasant to read.

Significance:
The results presented are novel to the best of my knowledge, and give some interesting insight on the behavior of Lloyd-like algorithms in semi-discrete OT problems.

Quality:
I think this paper is definitely a competent paper that should find its place at NeurIPS. It is somewhat frustrating that the paper does not deal with the case $\theta \mapsto W_2(T_\theta \circ \mu, \rho)$ as advertised in the introduction, though the current contributions of the paper are sufficient in my opinion. (note: the pushforward notation does not render properly so I use $\circ$ instead)

Questions for the authors:
- (a) The update considered in the paper ($Y_N \mapsto B_N(Y_N)$) requires to explicitly know the target **density** $\rho$. Why this is reasonable in some situation (typically in semi-discrete OT), one is more likely to face a sample $x_1,\dots, x_M \sim \rho$ (with $M$ large), or a discretization $\rho_M$ of $\rho$ as an histogram (as in Figure 3 for instance). Is it possible/easy to extend the results presented in the paper (in particular Corollary 4 and Theorem 7) to this (somewhat more general/realistic) setting? i.e. do we lose much by performing the update with respect to $x_1,\dots,x_M$ instead of $\rho$ directly when it comes to quantize $\rho$?
- (b) If my understanding is correct, Theorem 7 proves that $\mu_N = \delta_{Y_N^{k_N}}$ converges to $\rho$ for the metric $W_2$ (i.e. it is not "too bad"); but there is no guarantee that $\mu_N$ is (asymptotically) an optimal quantization of $\rho$. Do you have any clue on additional assumptions that may ensure that $F_N(\mu_N) /\min F_N \to 1$ (or even stronger, that $\mu_N$ is asymptotically optimal?)?

Minor remarks and suggestions:
- The notation $\delta_X$ for $\frac{1}{N} \sum_i \delta_{x_i}$ when $X = \{x_1,\dots,x_N\}$ may be a bit confusing, as $\delta_X$ refers to the measure supported on $\{x_1,\dots,x_N\}$ in $\mathbb{R}^{N \times D}$.
- (Figure 1 caption, typo?) The notation $[0;1]$ for intervals is not consistent with the notation $[0,1]$ used elsewhere in the paper.
- (Ref) It may be worth to mention [C, section 6.2] as another situation where people are interested in uniform quantization problems.
- (typo/notations) The notation $|P_i^{\frac{1}{2}}|$ has not been defined (I guess $| \cdot |$ is the diameter and the exponent $1/2$ should be out of the $| \cdot |$).
- (line 175) The naming ` ''Lloyd algorithm'' may be a bit misleading here as the paper considers a variant of this algorithm (the ``uniform'' Lloyd algorithm?).
- (Caption of Figure 2, typo) A $)$ is missing in $F_N ( \cdot )$
- Mention that proof of Corollary 6 can be found in the appendix.
- (Eq 22) Using $F_N(Y_N^0)$ would help to alleviate/harmonize notations (wrt the equation above).
- (line 237) ``"this figure'', a href is missing.
- (line 239) ``"red curve'', they seem blue (on my pdf reader at least).
- (related to question b) In [A,B], roughly speaking, authors give conditions on initialization so that the Lloyd algorithm converges toward an optimal (standard, not uniform) quantization of the target measure.

References:
- [A] Levrard 2015, _Nonasymptotic bounds for vector quantization in Hilbert spaces._
- [B] Chazal et al., 2020. _Clustering of measures via meanmeasure quantization._
- [C] Cuturi and Doucet, 2015, _Fast Computation of Wasserstein Barycenters._

Note:
 I checked the proofs of Theorem 3 and those of section 3; which seem correct as far as I can tell. The argument of using an orthogonal segment in the Minkowski sum to obtain the $d-1$ exponent may be detailed a bit. The proof of Corollary 5 was not checked.

**Time Spent Reviewing:**

10

---

> ### Author Response · Authors · 2021-08-10
> **Response to the first review**
>
> Dear Reviewer,
>
> We would like to start by thanking you for your thorough review and critique of our submission. We have tried to answer them in a satisfying fashion. Let us provide a more elaborate answer for some of them:
>
> > (a) The update considered in the paper ($Y_N \mapsto B_N(Y_N)$) requires to explicitly know the target density $\rho$. Why this is reasonable in some situation (typically in semi-discrete OT), one is more likely to face a sample $x_1,\dots, x_M \sim \rho$
> (with $M$ large), or a discretization $\rho_M$ of $\rho$ as an histogram (as in Figure 3 for instance).Is it possible/easy to extend the results presented in the paper (in particular Corollary 4 and Theorem 7) to this (somewhat more general/realistic) setting? i.e. do we lose much by performing the update with respect to $x_1,\dots,x_N$ instead $\rho$ of directly when it comes to quantize $\rho$?
>
> $\rightarrow$This is a good question that we haven't studied yet. It seems that one could leverage recent results on the stability of optimal transport maps (e.g. by Li-Nochetto) for this purpose.
>
> > (b) If my understanding is correct, Theorem 7 proves that $\mu_N = \delta_{Y_N^{k_N}}$ converges to $\rho$ for the metric $W_2$ (i.e. it is not "too bad"); but there is no guarantee that $\mu_N$ is (asymptotically) an optimal quantization of $\rho$. Do you have any clue on additional assumptions that may ensure that  (or even stronger, that $F_N(\mu_N) /\min F_N \to 1$ is asymptotically optimal?)?Do you have any clue on additional assumptions that may ensure that (or even stronger, that is asymptotically optimal?)
>
> $\rightarrow$Given the couter-example of Appendix E, it does not seem easy to find such assumption even in the simpler case of Corollary 4.
>
> Thank you again for noticing several typos, we will correct them. We will also try to integrate your suggestions to the final proposal.

---

> > ### Comment · Reviewer_nnEq · 2021-08-25
> > **Thanks**
> >
> > Thank you for your answer!

---

### Official Review · Reviewer_T7tc · 2021-07-15

**Rating:** 6
**Confidence:** 5

**Summary:**

In the paper, the author studies the problem of generating $N$ samples $Y_{1}, ..., Y_{N}$ such that they are the optimal solution of the following uniform quantization problem:
                                                    $$\inf_{Y \in \mathbb{\Omega}^{N}} W_{2}^{2}(P, \frac{1}{N} \sum_{i = 1}^{N} \delta_{Y_{i}}), \quad (1)$$
where $P$ is a given distribution on $\Omega$, $\Omega$ is a subset of $\mathbb{R}^{d}$. That problem can be thought as a constrained version of population K-means problem, which entails that
                                                    $$\inf_{G: |G| \leq N} W_{2}^2(P, G), \quad (2)$$
where $|G|$ indicates the number of supports of probability distribution $G$.

Since we only need to optimize the supports $Y_{1},..., Y_{N}$ from objective function (1), that optimization problem can therefore have a better optimization landscape than that of the population K-means problem in equation (2) where we need to optimize both the supports and weights of the probability measure $G$.

Based on that insight, the authors propose an adjusted version of Loyd's algorithm to solve objective function (1) and establish the convergence rate of that algorithm.

**Ethical Concerns:**

I do not detect any ethical issues with the paper.

**Ethics Review Area:**

["I don’t know"]

**Limitations And Societal Impact:**

I do not see any foreseeable limitations and negative societal impact of this work.

**Main Review:**

I think the viewpoint of considering a uniform quantization problem of generating the samples $Y_{1}, Y_{2}, ..., Y_{n}$ is of interest though some of the main results are based on previous results in the literature. The key point here is that we are able to obtain a favorable theoretical guarantee for Loyd's algorithm under that specific setting.

Here, I have the following comments with the paper:

(1) There are typos in Theorem in page 4. The condition $\|y_{i} - y_{j}\| \geq c N^{-1/d}$ should be changed to $\|y_{i}^{0} - y_{j}^{0}\| \geq c N^{-1/d}$.

(2) I wonder whether the assumption that the parameter space $\Omega$ is compact and convex can be further relaxed? One key insight from the K-means problem (2) is that even though the supports of probability measure $G$ lie on $\mathbb{R}^{d}$, eventually we can find a ball of some radius such that these supports will lie on. I think that a similar viewpoint may also hold for the uniform quantization problem that the authors consider.

(3) In practice, what is good guidance for the choice of $N$? For example, if the distribution $P$ in equation (1) is a mixture of Gaussian distributions, poor choice of $N$ may lead to the lack of samples in some clusters or a very large value of $N$ may lead to the redundancy of samples in some clusters. Furthermore, the fact that the error rate is proportional to $N^{-1/d}$ is quite painful for practical application as we need to choose $N$ to be proportional to $d$ and $d$ is usually very large in practice.

(4) In Proposition 2, do we have convergence speed for some sub-sequences of $\{F_{N}(Y^{k})\}$ to critical points? Furthermore, will $F_{N}$ be locally strongly convex around the minimizes?

(5) In the proof of Theorem 3, several places used $I_{\epsilon}$. The authors should change it to $I_{\epsilon}(Y)$. Furthermore, I may misunderstand the statement of Theorem 3. If we allow $\epsilon$ goes to 0 in the RHS of equation (15) in Theorem 3, the RHS will go to 0. On the other hand, the LHS does not depend on $\epsilon$, which means its value will be 0 as $\epsilon$ goes to 0. I feel that there is a typo with the assumption of $Y$ in that Theorem 3, namely, $Y \in \Omega^{N} \backslash D_{N, \epsilon}$. The authors please check that potential typo.

(6) The experimental results are quite poor as they are mainly in 2D. The authors may need to carry out more experiments in high-dimensional settings to showcase the benefits of their methods.

(7) The authors may consider adding a few relevant machine learning references using optimal transport for quantization purposes, such as clustering, which have quite a similar flavor as that being studied in the paper. A few representative examples include: multilevel clustering via optimal transport [1],  approximate optimal transport using quantization [2].

(8) Some other typos:

- Line 190: "It is possible for to" -> "It is possible for us to".

- In Corollary 4, I think we should change $D_{\epsilon_{N}}$ to $D_{N, \epsilon_{N}}$.

--- References:

[1] N. Ho, L. Nguyen, M. Yurochkin, H. Bui, V. Huynh, and D. Phung. Multilevel clustering via Wasserstein means. ICML, 2017.

[2] G. Beugnot, A. Genevay, K. Greenewald, J. Solomon. Improving Approximate Optimal Transport Distances using Quantization. UAI, 2021.

**Time Spent Reviewing:**

4 hours

---

> ### Author Response · Authors · 2021-08-10
> **Response to the first review**
>
> Dear Reviewer,
>
> We would like to start by thanking you for your thorough review and critique of our submission. We have tried to answer them in a satisfying fashion. We list here the ones which, we thought needed a more elaborate answer:
>
> > (2) I wonder whether the assumption that the parameter space  is compact and convex can be further relaxed? One key insight from the K-means problem (2) is that even though the supports of probability measure  lie on , eventually we can find a ball of some radius such that these supports will lie on. I think that a similar viewpoint may also hold for the uniform quantization problem that the authors consider.
>
> $\rightarrow$ We note that the support of the measures does not need to be convex ; however, the estimates in our main results depend on the diameter of the support (or, equivalently,
> of its convex hull $\Omega$). In practice, our main theorem applies with slight adaptation (in the definition of the Laguerre cells and barycenters) to e.g. a finitely
> supported rho. On the other hand, it does not seem obvious how to discard the compactness hypothesis. We will highlight these remarks more clearly.
>
> > (3) In practice, what is good guidance for the choice of $N$? For example, if the distribution $P$ in equation (1) is a mixture of Gaussian distributions, poor choice of $N$ may lead to the lack of samples in some clusters or a very large value of $N$ may lead to the redundancy of samples in some clusters. Furthermore, the fact that the error rate is proportional to $N^{-1/d}$ is quite painful for practical application as we need to choose $N$ to be proportional to $d$ and $d$ is usually very large in practice.
>
> $\rightarrow$ We agree, but this exponent comes from the fact that we assume that our measure rho is absolutely continuous wrt the Lebesgue measure (in which case the minimum quantization error
> is $O((1/N)^{2/d})$ anyway). One could hope that if $\rho$ is the volume measure over a $k$-dimensional submanifold (perhaps with additional assumptions, such as upper bounds on its curvature),
> then the error rate will behave as in the k-dimensional case: this will be the object of further studies.
>
> > (4) In Proposition 2, do we have convergence speed for some sub-sequences of $F_N(Y^k)$ to critical points? Furthermore, will $F_N$ be locally strongly convex around the minimizers?
>
> $\rightarrow$ We do not have convergence speed results for Lloyd's algorithm for uniform quantization. Such result seem hard to obtain since the cluster points might be non-minimizing critical points. Actually,
> very little is known about the critical points of $F_N$: there exists critical points with high energy, but we do not have any example of *stable* critical point with high energy.
>
> > (5) In the proof of Theorem 3, several places used $I_\epsilon$. The authors should change it to $I_\epsilon(Y)$. Furthermore, I may misunderstand the statement of Theorem 3. If we allow $\epsilon$ goes to 0 in the RHS of equation (15) in Theorem 3, the RHS will go to 0. On the other hand, the LHS does not depend on $\epsilon$, which means its value will be 0 as $\epsilon$ goes to 0. I feel that there is a typo with the assumption of Y in that Theorem 3, namely, $Y\in\Omega^N\setminus\mathbb{D}_{N,\epsilon}$. The authors please check that potential typo.
>
> $\rightarrow$ We do not believe that there is a typo (but we will check carefully): note that the exponent of $\varepsilon$ in the upper bound is negative, so that it goes to $+\infty$ when $\varepsilon$ goes to
> $0$ (so that there is indeed a trade-off to make in order to choose $\varepsilon$ in order to optimize the upper bound).
>
> > (6) The experimental results are quite poor as they are mainly in 2D. The authors may need to carry out more experiments in high-dimensional settings to showcase the benefits of their methods.
>
> $\rightarrow$ Our experiments are done in a low-dimensional setting, where we can resort to quadratures for computing the barycenters (we rely on the PySDOT library for this purpose). In higher dimensions, this could be done if the source measure is finitely supported (in which case barycenters are simply weighted averages), or using Monte-Carlo methods.
>
> We will correct the typos mentioned in your review as well as add some references to similar problems in machine learning.

---

> > ### Comment · Reviewer_T7tc · 2021-08-25
> > **Response to the authors**
> >
> > I would like to thank the authors for spending time responding to my reviews. The responses of the authors address some of my concerns; therefore, I decide to keep my current score.

---

### Official Review · Reviewer_gM3z · 2021-07-17

**Rating:** 7
**Confidence:** 3

**Summary:**

This paper studies Lloyd's algorithm and gradient descent for Wasserstein regression and provides approximation and convergence guarantees.

**Limitations And Societal Impact:**

This is a theoretical paper, and I think there is no potential negative societal impact.

**Main Review:**

Strengths:
1. This paper tackles the problem of approximating a model distribution using a pushforward of a discrete reference distribution, which is hard due to non-convexity. The authors establish some theoretical guarantees for one-step Lloyd's algorithm and gradient descent, showing that these methods are valid and useful for the problem.
2. The theoretical results are presented clearly, and numerical experiments are conducted to verify the theory.

Weakness / comments:
1. The non-asymptotic convergence rate in Corollary 4, 5 and 7 does not match the upper bound on the minimum of $F_N$ in Remark 1. It might be beneficial to explain why the present theory is not sharp, and which part of the analysis is not tight.
1. Corollary 4 present the results for running Lloyd's algorithm for only one step. Will it get improved if we run more steps? Is it possible to get an error rate that matches the rate in Remark 1? (either from theoretical perspective or numerical perspective)
2. It would be more convincing to include the case when $d\geq3$ in the numerical experiment.

**Time Spent Reviewing:**

2

---

> ### Author Response · Authors · 2021-08-10
> **Response to the first review**
>
> Dear Reviewer,
>
> First, we would like to thank you for your detailed review of our submission. We have tried to answer your critiques in a satisfying way:
>
> You noticed that "The non-asymptotic convergence rate in Corollary 4, 5 and 7 does not match the upper bound on the minimum of
> in Remark 1. It might be beneficial to explain why the present theory is not sharp, and which part of the analysis is not tight".
>
> $\rightarrow$ Actually, the analysis of Corollary 4 is tight when $\beta=d$, this is explained in Remark 3 and Appendix E. The minimum quantization error is
> indeed $O((1/N)^{2/d}$) (Remark 1). However, for any $\alpha>1/d$, it is possible to construct examples of densities rho and point clouds $Y_N$ (such that the minimal
> distance between points is maximal, i.e. than $O((1/N)^{1/d})$ such that the quantization error after one step of Lloyd's algorithm is $O((1/N)^{\alpha})$. This
> is explained in Corollary 5 of the appendix. This proves that the exponent in Corollary 4 cannot be improved when $\beta=d$. We will make this optimality more
> prominent in the final version by moving the statement of Corollary 5 to the main text.
>
> Another remark was that "Corollary 4 presents the results for running Lloyd's algorithm for only one step. Will it get improved if we run more steps? Is it possible to get an error rate that matches the rate in Remark 1? (either from theoretical perspective or numerical perspective)".
>
> $\rightarrow$ We assume here that you are referring to Remark 2, which gave theoretical decay rates for the minimum values of $F_N$. First let us notice that in the cases described in Appendix E, Corollary 9, it is not possible to match the error rate of Remark 1 even with more steps of Lloyd's algorithm. Indeed, this counter-example yields a critical
> point of Lloyd's algorithm after one step of Lloyd's algorithm, so that further iterations will not change the positions of the Dirac masses.
>
> Finally, you made a remark that "It would be more convincing to include the case when $d\geq3$ in the numerical experiment.
>
> $\rightarrow$ Our experiments are done in a low-dimensional setting, where we can resort to quadratures for computing the barycenters (we rely on the PySDOT library for this purpose). In higher dimensions, this could be done if the source measure is finitely supported (in which case barycenters are simply weighted averages), or using Monte-Carlo methods.

---

> > ### Comment · Reviewer_gM3z · 2021-08-27
> > **Response**
> >
> > Thank you for taking time responding to my reviews. I've updated my score accordingly.

---

### Decision · Program_Chairs · 2021-09-28

**Decision:**

Accept (Poster)

**Comment:**

All the reviewers stressed that this paper presents strong theoretical and numerical contributions, and it should be accepted.  As a side note, I think the authors should cite the paper “A note on constrained k-means algorithms, Michael K. Ng, 2000” which proposed the idea of quantizing toward uniform weights. The paper “Fast Computation of Wasserstein Barycenters Marco Cuturi, Arnaud Doucet, 2013” also contains an extensive discussions and numerical tests about this approach.

**Consistency Experiment:**

NeurIPS has a long history of experimentation. In 2014, NeurIPS ran an experiment in which 10% of submissions were reviewed by two independent committees to quantify the randomness in the review process. This year, we repeated a variant of this experiment to see how the quality of the review process has changed over time.  This paper was part of the experiment and was therefore assigned to two committees (consisting of reviewers, an Area Chair, and a Senior Area Chair) that reached independent decisions.  If both committees made the same recommendation, this recommendation was followed. If a single committee recommended acceptance, the paper was accepted (with the exception of a few cases in which the other committee identified what we considered a fatal flaw, e.g., an error in a key result).

This copy’s committee reached the following decision: **Accept (Spotlight)**

The other committee assigned to the paper recommended **Reject**.  You can find the other set of reviews, along with any follow up discussion with the authors here:
https://openreview.net/forum?id=_6j_jQiYB2c